# Fumarate Hydratase Enhances the Therapeutic Effect of PD-1 Antibody in Colorectal Cancer by Regulating PCSK9

**DOI:** 10.3390/cancers16040713

**Published:** 2024-02-08

**Authors:** Le Qin, Liang Shi, Yu Wang, Haixin Yu, Zhouyuan Du, Mian Chen, Yuxuan Cai, Yinghao Cao, Shenghe Deng, Jun Wang, Denglong Cheng, Yixin Heng, Jiaxin Xu, Kailin Cai, Ke Wu

**Affiliations:** 1Department of Gastrointestinal Surgery, Tongji Medical College, Union Hospital, Huazhong University of Science and Technology, Wuhan 430022, China; qinle_xj@163.com (L.Q.); lianguh@126.com (L.S.); chenmian1997@163.com (M.C.); josephsimmonscyx@163.com (Y.C.); dengshenghe@hust.edu.cn (S.D.); wangjun@foxmail.com (J.W.); chengdenglong9704@163.com (D.C.); 2Department of General Surgery, The First Affiliated Hospital of Shihezi University, Shihezi 832008, China; hyxwztgd@163.com (Y.H.); gtr0101@163.com (J.X.); 3Department of Gastrointestinal Surgery, The Second Affiliated Hospital of Zhejiang University School of Medicine, Hangzhou 310009, China; yuwanggs@zju.edu.cn; 4Department of Digestive Surgical Oncology, Cancer Center, Tongji Medical College, Union Hospital, Huazhong University of Science and Technology, Wuhan 430022, China; haixin.y@outlook.com (H.Y.); duzhouyuan@gmail.com (Z.D.); 2022xh0021@hust.edu.cn (Y.C.)

**Keywords:** colorectal cancer, fumarate hydratase, immunotherapy, programmed cell death 1, protein invertase subtilisin/kexin 9 type

## Abstract

**Simple Summary:**

This research revealed that the downregulation of fumarate hydratase in patients is associated with poor prognosis. Mechanistically, FH binds to RAN, which inhibits the nuclear import of the PCSK9 transcription factor SREBF1/2, thus reducing the expression of PCSK9. This leads to increased clonal expansion of CD8+ T cells while the number of Tregs remains unchanged, and the expression of PD-L1 does not change significantly, thus enhancing the immunotherapy response. Importantly, combined therapy targeting PCSK9 and PD-1 may be beneficial for patients with CRC and low FH expression. Considering these results, our findings hold potential for future clinical applications.

**Abstract:**

Despite the notable achievements of programmed death 1 (PD-1) antibodies in treating various cancers, the overall efficacy remains limited in the majority of colorectal cancer (CRC) cases. Metabolism reprogramming of tumors inhibits the tricarboxylic acid (TCA) cycle, leading to down-regulation of fumarate hydratase (FH), which is related to poor prognosis in CRC patients. By establishing a tumor-bearing mouse model of CRC with Fh1 expression deficiency, we confirmed that the therapeutic effect of PD-1 antibodies alone was suboptimal in mice with low Fh1 expression, which was improved by combination with a protein invertase subtilisin/kexin 9 (PCSK9) inhibitor. Mechanistically, FH binds to Ras-related nucleoprotein (RAN), which inhibits the nuclear import of the PCSK9 transcription factor SREBF1/2, thus reducing the expression of PCSK9. This leads to increased clonal expansion of CD8+ T cells while the number of Tregs remains unchanged, and the expression of PD-L1 does not change significantly, thus enhancing the immunotherapy response. On the contrary, the expression of PCSK9 increased in CRC cells with low FH expression, which antagonized the effects of immunotherapy. Overall, CRC patients with low FH expression may benefit from combinatorial therapy with PD-1 antibodies and PCSK9 inhibitors to enhance the curative effect.

## 1. Introduction

Colorectal cancer (CRC) is the second leading cause of cancer-related death in humans [1]. In some patients diagnosed with advanced CRC, the 5-year survival rate is only 14% [2]. Compared with conventional chemotherapy and targeted therapy, immunotherapy has changed the treatment prospects for patients with various solid tumors and has become the standard regimen in CRC [3]. Programmed cell death 1 (PD-1) antibodies, serving as immune checkpoint inhibitors (ICIs), have demonstrated notable therapeutic efficacy in patients exhibiting high microsatellite instability (MSI-H), a condition present in approximately 15% of all CRC cases [4]. Regrettably, the predominant subset of CRC patients characterized by microsatellite stability (MSS) typically shows a lack of response to ICI therapy. In 2018, only 12.5% of patients with cancer who received treatment with ICIs in the United States had an objective response [5]. This may be due to inter-patient differences in the tumor microenvironment (TME) and differences in tumor cell clonality, which reflects the highly regulated and complex nature of the immune system [6,7]. Therefore, understanding the inhibitory mechanisms of immunotherapy in TME will facilitate the development of personalized immunotherapy regimens, thereby improving the efficacy of tumor immunotherapy.

Metabolic disorder is not only the result of carcinogenic transformation but also one of the main driving factors in cancer development. Fumarate hydratase (FH) is an important metabolic enzyme in the tricarboxylic acid (TCA) cycle, and the loss of FH can act as a cancer driver [8]. In addition, FH plays a protective role in maintaining the interaction between interferon and cytokines in macrophages [9]. This indicates that there may be a relation between FH expression and the efficacy of tumor immunotherapy; however, the specific mechanism remains unclear. We found that the expression of FH was lower in some of the CRC patients. Moreover, through RNA-seq, we identified that protein invertase subtilisin/kexin 9 (PCSK9) may be a downstream effector of FH. Importantly, PCSK9 can reduce the expression of major histocompatibility complex (MHC) I protein on the surface of tumor cells, weaken T-cell receptor circulation and signal transmission, and inhibit the tumor infiltration and anti-tumor activity of CD8+ T cells [10,11].

Our results showed that low FH expression was related to increased Ras-related nucleoprotein (RAN)-mediated nuclear translocation of the PCSK9 transcription factor, SREBF1/2, which promoted the expression of PCSK9, thus weakening the monotherapy effect of a PD-1 antibody. Moreover, PCSK9 inhibition combined with PD-1 antibody therapy enhanced the effect of tumor immunotherapy. Therefore, we propose that FH may regulate sensitivity to treatment with PD-1 antibodies in patients with CRC and that patients with CRC with low FH expression may benefit from combined treatment with PCSK9 inhibitors and PD-1 antibodies. In addition, targeting key molecular entities in tumors can help maximize therapeutic efficacy [12,13], and FH has the potential in this regard.

## 2. Materials and Methods

### 2.1. Data Mining and Bioinformatics Analysis

The Cancer Genome Atlas (TCGA), UCSC Xena platform (UCSC Xena), Gene Set Enrichment Analysis (GSEA), and ChIP-Atlas were used for data mining and bioinformatics analysis (please refer to Appendix A for details).

### 2.2. Clinical Specimen

We collected postoperative cancer tissue and adjacent normal tissue from 12 patients with CRC who were treated at the Union Hospital Affiliated with Tongji Medical College of Huazhong University of Science and Technology. Postoperative pathological specimens from six rectal cancer patients were sensitive to neoadjuvant radiotherapy and chemotherapy combined with PD-1 antibodies (defined as tumor shrinkage of ≥40%), and six patients were insensitive (defined as tumor shrinkage < 40%). All procedures in this study were approved by the Ethics Committee of Union Medical College, Affiliated with Tongji Medical College, Huazhong University of Science and Technology (2022IEC-094). Prior to this study, all patients provided written informed consent. CRC tissue microarrays (HCol-Ade060CS-01,60 cases) were purchased from Shanghai Xinchao Biotechnology Co., Ltd. (Shanghai, China). The research methods used conform to the standards stipulated in the Helsinki Declaration.

### 2.3. Cell Line and Culture

A normal colon epithelial cell line (HCoEpiC), several human colon cancer cell lines (HCT15, LoVo, HCT116, SW620, SW480), and a mouse colon cancer cell line (MC38) were purchased from Type Culture Collection Cell Bank, Chinese Academy of Sciences (Shanghai, China). The authenticity of all cell lines was confirmed by short tandem repeat (STR) DNA analysis. HCT15 was cultured in RPMI-1640 medium (Gibco, Cat No. 11875093, Grand Island, NY, USA). LoVo was cultured in an F-12K medium (Gibco, Cat No. 21127022, Grand Island, NY, USA). HCT116 was cultured in McCoy’s 5A medium (Gibco, Cat No. 16600082, Grand Island, NY, USA). HCoEpiCS, SW620, SW480, and MC38 were cultured in high glucose DMEM medium (Gibco, Cat No. C11995500BT, Grand Island, NY, USA). All cells were cultured in 10% FBS (CELLiGENT, Cat No. CG0430B, Hamilton, New Zealand) and a 1% mixed solution of penicillin and streptomycin (Gibco, Cat No. 15140122, Toronto, ON, Canada). Cell lines were cultured in a saturated humidity incubator at 37 °C and 5% CO_2_.

### 2.4. Cell Transfection and RNA Interference Based on Lentivirus

An FH overexpression plasmid and empty vector (pcDNA3.1) was purchased from OriGene Company (Cat No. RC200614, Rockville, MD, USA). Lipofectamine 2000 (Invitrogen, Cat No. 11668019, Waltham, MA, USA) was used to transfect plasmids. FH-specific lentivirus (shRAN) was purchased from Gene-Chem (Shanghai, China). At 16 h post-transfection, the medium was changed, and 48 h later, the cell lines were screened for knockdown of the FH gene using 4 μg/mL puromycin (Bioroxx, Cat No. 1299MG025, Nordrhein-Westfalen, Germany) or 100 μg/mL Hygromycin B (BioFroxx, Cat No. 1366ML010, Nordrhein-Westfalen, Germany). The effects of plasmid transfection and lentivirus infection were verified by Western blot or RT-qPCR. The sequences of all shRNA constructs are listed in Appendix A.

### 2.5. Western Blot Analysis and Co-Immunoprecipitation (Co-IP)

Cell or tissue proteins were collected, and total proteins were extracted using RIPA buffer (Thermo Scientific, Cat No. 89900, Waltham, MA, USA) containing protease inhibitors (Bimake, Cat No. B14001, Houston, TX, USA). Cytoplasmic protein and nuclear proteins were extracted using a Nuclear and Cytoplasmic Protein Extraction Kit (Beyotime, Cat No. P0028, Shanghai, China). A BCA protein concentration determination kit (Beyotime, Cat No. P0012, Shanghai, China) was used to determine the protein concentration. Proteins were loaded onto SDS/PAGE gels for electrophoresis and transferred to PVDF membranes (Millipore, Cat No. IPVH00010, Burlington, MA, USA). Following overnight incubation with the primary antibody, membranes were incubated with the respective secondary antibody and detected by enhanced chemiluminescence (ECL). ImageJ (Version: 1.50G) software was used for protein quantitative analysis. The normalization with beta actin has been done in new clean blots (in general), and in a few cases in the same blot.

For co-immunoprecipitation, cells were lysed with IP lysis buffer (Servicebio, Cat No. G2038-100ML, Wuhan, China) containing phosphorylase inhibitor (Servicebio, Cat No. G2007-1ML, Wuhan, China) and PMSF (Servicebio, Cat No. G2008-1ML, Wuhan, China). The cell lysate was centrifuged at 12,000 rpm for 25 min. Next, the supernatant, corresponding antibody, and protein A/G agarose beads (Santa Cruz, Cat No. sc-2003, Santa Cruz, CA, USA) were incubated overnight at 4 °C on a shaker. The immune complex was washed with IP lysis buffer mixed with loading buffer, and immunoblotting was carried out. The antibodies used are listed in Appendix A.

### 2.6. Quantitative Real-Time PCR Analysis

Total RNA was extracted with TRIzol (Takara, Cat No. T9108, Shiga, Japan) according to the manufacturer’s instructions. cDNA was synthesized by reverse transcription with HiScript II Q RT SuperMix for qPCR kit (Vazyme, Cat No. R222-01, Nanjing, China), and RT-qPCR was performed with Chamq Universal Sybr QCPR Master Mix kit (Vazyme, Cat No. Q711-02, Nanjing, China). β-Actin was used as a reference, and the calculation formula is 2^−ΔΔCT^. The primers used are listed in Appendix A.

### 2.7. Transwell Invasion Assay

The Transwell invasion assays were conducted using Transwell cell culture chambers (Corning, Cat No. 3422, Corning, NY, USA) with a diameter of 6.5 mm and a pore size of 8 μm. A total volume of 300 µL serum-free Opti-MEM (Gibco, Cat No. 31985070, Grand Island, NE, USA) was seeded into the upper chamber coated with Matrigel (BD Bioscience, Cat No. 356234, Beijing, China) at a concentration of 50 μL/cm^2^. Simultaneously, 600 µL medium containing 10% FBS was added to the lower chamber. After incubation for 24 h, cells were fixed with 4% paraformaldehyde (Servicebio, Cat No. G1101, Wuhan, China), followed by staining with a 1% crystal violet solution. Images were captured under an inverted microscope after wiping the upper surface of the chamber with sterile cotton swabs. Finally, the cells that had migrated through polycarbonate (PC) membranes were analyzed using Image J software (Version: 1.50G).

### 2.8. Colony-Forming Assay

1 × 10^3^ HCT116 or SW620 cells were seeded into 6-well tissue culture plates following different treatments. Following a 14-day culture, the cells were fixed and stained using a 1% crystal violet solution, and visible colonies were observed through counting.

### 2.9. CCK8 Assay

The CCK8 assay was performed according to the manufacturer’s instructions. Briefly, a Cell Proliferation Assay Cocktail Kit (Abbkine, Cat No. KTD103-CN, Wuhan, China) was used to analyze cell proliferation. A total of 5 × 10^3^ cells/well containing 100 μL of medium were seeded in 96-well plates. The cells were incubated in a humidified incubator at 37 °C with 5% CO_2_ for 24, 48, 72, 96, or 120 h, and 10 μL CCK-8 solution was added at the corresponding time points and cultured in the incubator for another 2 h. Wells without cells were used as blank controls, and absorbance at 450 nm was measured using a microplate reader (PerkinElmer EnSpire, SG, Pleasanton, CA, USA).

### 2.10. Flow Cytometry Analysis

In total, 100 μL of MC38 cell suspension (approximately 2 × 10^6^ cells) corresponding to the shFh1Control or shFh1 was inoculated subcutaneously into C57BL/6 mice. Tumors were collected on day 21 after inoculation, minced and incubated in DNase I (BioFroxx, 0.2 mg/mL, Cat No. 1121MG010, Einhausen, Germany), collagenase I (BioFroxx, 1 mg/mL, Cat No. 1904MG100, Einhausen, Germany) and dispase II (Sigma-Aldrich, 2 mg/mL, Cat No. D4693, Burligton, MA, USA) for 60 min at 37 °C in a constant temperature shaker. After termination of digestion, the cells were filtered through a 70 μm cell filter to obtain a single-cell suspension. Cells were incubated with antibodies targeting cell surface antigens for 30 min at 4 °C in the dark using the following antibodies: CD45 antibody (BD, FITC, Cat No. 553080, Franklin Lake, NJ, USA), CD3 antibody (BioLegend, PE/Cyanine7, Cat No. 100219, San Diego, CA, USA), CD4 antibody (BioLegend, APC/Cyanine7, Cat No. 100525, San Diego, CA, USA), CD8a (BioLegend, PerCP/Cyanine5.5, Cat No. 100733, San Diego, CA, USA) and PD-L1 antibody (BioLegend, Biotin, Cat No. 124305, San Diego, CA, USA). Cells were fixed/permeabilized with 1× Foxp3 Fix/Perm Buffer (Invitrogen, Cat No. 2518973, Carlsbad, CA, USA) and incubated with Foxp3 antibody (Invitrogen, Cat No. 2518973, California, CA, USA) or isotype control IgG1 antibody (BioLegend, PE, Cat No. 400139, San Diego, CA, USA) in 1× Foxp3 Perm Buffer (Invitrogen, Cat No. 2518973, Carlsbad, CA, USA) for 30 min at room temperature in the dark. Finally, analyses were performed using a BD FACSCanto II flow cytometer. FlowJo (V10) was used to analyze the data.

### 2.11. Immunohistochemistry (IHC)

Human or mouse subcutaneous tumor model CRC tissue specimens were fixed with 4% paraformaldehyde and embedded in paraffin to prepare 4-μm sections. The prepared tissue sections or tissue microarrays were decaffinized and rinsed with water. The tissue antigens were repaired with citric acid repair solution (PH 6.0), and endogenous enzymes were blocked with 3% H_2_O_2_. After blocking with serum from the same source as the secondary antibody, the primary antibody was incubated overnight at 4 °C and the secondary antibody at 37 °C for 1 h. Diamino-benzidine (DAB) color solution (Servicebio, Cat No. G1212-200T, Wuhan, China) was used for color development, followed by counterstaining with hematoxylin, dehydration, and sealing. Images were obtained after microscopic observation. The primary antibodies used are listed in Appendix A. The pathologist scored the staining using a double-blind method. The IHC score was determined as the percentage of positive cells multiplied by the intensity of staining (0–3).

### 2.12. Animal Studies

Mice were provided by Hunan Westlake Jingda Laboratory Animal Co., Ltd. (Changsha, China), and the animal experimental protocol was approved by the Animal Ethics Committee of Huazhong University of Science and Technology and conducted accordingly (S3563). Mice were randomly assigned to the indicated groups. Tumor volumes were calculated according to the formula (L × W2)/2. HCT116 cells were infected with shControl, shFH, shPCSK9, or shFH+PCSK9 lentivirus after stable screening. Nude mice were subcutaneously inoculated with 100 μL of HCT116 cell suspension corresponding to approximately 2 × 10^6^ cells to establish a subcutaneous tumor model. Mouse MC38 cell lines infected with shControl and shFh1 lentiviruses were stably screened. C57BL/6 mice were subcutaneously inoculated with 100 μL of MC38 cell suspension to establish a C57BL/6 mouse subcutaneous tumor model. PD-1 antibody (Bioxcell, Cat No. BP0146, Lebanon, NH, USA) and isotype IgG antibody (Bioxcell, Cat No. BE0089, Lebanon, NH, USA) against MC38 subcutaneous tumor models were administered by intraperitoneal injection at a dose of 5 mg/kg every 3 days. A PCSK9 inhibitor (Selleck, Cat No. 489415-96-5, Houston, TX, USA) and DMSO were administered subcutaneously at a dose of 4 mg/kg/day.

### 2.13. RNA Sequencing and Analysis

Total RNA was extracted from cells using TRIzol (Takara, Cat No. T9108, Japan). RNA sequencing (RNA-Seq) was performed by Novogene (Beijing, China). RNA integrity was verified using an Agilent 2100 bioanalyzer (Agilent Technologies, Santa Clara, CA, USA). Double-ended library sequencing on Illumina HiSeq 2500 was performed with the mRNA-seq sample preparation kit (Illumina, San Duego, CA, USA) according to the instructions provided by the manufacturer. Next, the sequencing data were analyzed using the Illumina data analysis pipeline. To minimize experimental bias, all samples were assigned to lane assignments and performed in a blinded fashion.

### 2.14. Chromatin Immunoprecipitation (ChIP) and ChIP-qPCR

Follow the manufacturer’s instructions. ChIP assays were performed using the SimpleChIP Enzymatic Chromatin IP Kit (Cell Signaling Technologies, Danvers, MA, USA, Cat No. 9003). After washing and purification, the DNA was analyzed by qPCR. Specific antibodies are shown in Appendix A, and primer sequences are shown in Appendix A.

### 2.15. Immunofluorescence Staining

Cells were inoculated in a 12-well plate and fixed with 4% paraformaldehyde (Servicebio, Cat No. G1101, Wuhan, China) for 20 min. After washing, cells were sealed with 5% bovine serum albumin (Sigma-Aldrich, Cat No. A1933, USA) for 1 h, followed by incubation overnight with primary antibody at 4 °C (see Appendix A for specific antibodies). Next, cells were washed three times and incubated with CoraLite 594-Conjugated Goat Anti-Rabbit IgG (H+L) (Protein Tech, Cat No. SA00013-4, 1:100 dilution, Wuhan, China) at room temperature for 1 h. The nuclei were stained with DAPI (Beyotime, Cat No. C1006, Shanghai, China) and incubated at room temperature for 10 min. Anti-fluorescence quenching sealing solution (Beyotime, Cat No. P0126, Shanghai, China) was used to seal the film for 5 min. Images were taken under a fluorescence microscope (Olympus IX73, Tokyo, Japan).

### 2.16. Statistical Analysis

GraphPad Prism 9.4 software and R 4.3.0 software were used for statistical analysis; experimental data are expressed as mean ± SD, and at least three independent experiments were repeated in all cases. Unpaired two-tailed Student’s *t*-test or Mann-Whitney U test was used to compare the differences between the two groups and a two-way ANOVA test was used to compare differences between the two groups in the presence of time factors. Differences were considered statistically significant when *p*-values were less than 0.05. Survival curves were plotted by Kaplan-Meier analysis and assessed by log-rank test (* *p* < 0.05, ** *p* < 0.01, *** *p* < 0.001; NS indicates no statistical difference).

## 3. Results

### 3.1. FH Expression Is Downregulated and Related to Poor Prognosis in CRC

For cancer cells, the increased metabolic flux through glycolysis and suppression of the TCA cycle has been demonstrated to provide biosynthetic precursors for rapid macromolecule synthesis and to maintain cellular redox homeostasis for better survival [14]. However, the effect of metabolite accumulation on CRC due to inhibition of the TCA cycle has been little studied. We analyzed the differential gene expression between CRC tissues and normal colorectal tissue in the TCGA TARGET GTEx dataset, followed by gene set enrichment analysis (GSEA). The results of GSEA in the TCGA TARGET GTEx dataset showed that the TCA cycle pathway was significantly down-regulated in CRC tissues (Figure 1A). Next, we focused on the top 15 genes in the TCA genes from the TCGA TARGET GTEx dataset (Figure 1B) and found that the expression of succinate coenzyme A ligase (SUCLG2), succinate dehydrogenase B (SDHB) and fumarate hydratase (FH) significantly improved the prognosis of colon adenocarcinoma (COAD) in the TCGA cohort (Figure 1C). In the TCGA cohort, these TCA genes have the same protective trend on the prognosis of rectal adenocarcinoma (READ), but there is no statistical significance (Figure 1D). The mRNA expression levels of FH were significantly lower in tumor tissues than in adjacent normal tissues, according to TIMER 2.0 (Figure 1E). Our further experimental results showed that protein and mRNA expression levels of FH in tumor tissues were significantly lower than in normal tissues (Figure 1F,G). Considering the significant expression difference of FH in previous studies and its impact on CRC prognosis [15,16], we chose FH for further investigation.

### 3.2. FH Inhibited the Proliferation and Invasion of CRC Cells

In order to investigate the effect of FH expression on cancer cells, we first checked the expression of FH in various CRC cell lines to choose the optimal cell lines used for silencing and overexpression of FH. Our results showed that FH expression was higher in HCT116 and SW620 cell lines; therefore, these cell lines were selected for subsequent experiments (Figure 2A). To determine the biological function of FH in CRC cells, the stable FH knockdown cell lines were established by using lentivirus transfection. The FH expression of cell lines was verified by Western Blot and quantitative real-time polymerase chain reaction (RT-qPCR) (Figure 2B,C). We evaluated the growth characteristics of these CRC cells by performing colony formation and CCK-8 assays, and the results showed that FH knockdown promoted colony formation and proliferation in HCT116 and SW620 cells compared with controls (Figure 2D–G). Moreover, the proliferative capacity of the cells was inhibited after re-overexpression of FH when FH was knocked down (Appendix A). Our Transwell assay showed that the knockdown of FH promoted the invasion of HCT116 and SW620 cells (Figure 2H,I). We then generated cell lines with stable FH overexpression, which were confirmed by Western Blot and RT-qPCR (Figure 2J,K). In contrast, overexpression of FH reduced colony formation and proliferation of HCT116 and SW620 cells (Figure 2L–O). In addition, overexpression of FH inhibited the invasion of HCT116 and SW620 cells, as shown by our Transwell assay (Figure 2P,Q). These findings suggest that low expression of FH may promote the proliferation and invasion process of CRC cells.

### 3.3. Low Expression of FH Can Inhibit the Clonal Expansion of CD8+ T Cells in CRC and Reduce the Therapeutic Effect of PD-1 Antibodies

To investigate whether FH has an impact on the therapeutic efficacy of CRC, we collected samples from 12 patients with rectal cancers treated with neoadjuvant chemoradiotherapy combined with PD-1 antibodies, of which six were treatment-sensitive (tumor reduction of more than 40%) and 6 were treatment-insensitive (tumor reduction of less than 40%). We validated FH expression using Western Blot and found that FH expression was significantly increased in the treatment-sensitive samples (Figure 3A,B). This suggests that low FH expression may affect PD-1 antibody treatment efficacy. Therefore, we constructed an Fh1 stable knockdown mouse model using the mouse colon cancer cell line MC38. A C57BL/6 mouse model bearing subcutaneous CRC tumors was established to observe the effects of PD-1 antibody treatment on tumor growth. Our in vivo results showed that knockdown of Fh1 reduced immunotherapy efficacy and led to insignificant tumor volume regression (Figure 3C–E). To further explore the possible mechanisms by which FH affects therapeutic efficacy, we analyzed the expression of CD8, FOXP3, and PD-L1 in the subcutaneous tumor tissue of Fh1 knockdown mice by flow cytometry. Our findings showed that the expression of CD8 was significantly downregulated in subcutaneous tumor tissues with Fh1 knockdown (Figure 3F), while FOXP3 and PD-L1 were not significantly changed (Figure 3G,H). This suggests that the beneficial effect of Fh1 on immunotherapy efficacy may be achieved by affecting CD8+ T cells. To further test this, we evaluated the expression of PD-L1, CD8, and FOXP3 in CRC tissues by immunohistochemistry. The results showed that Fh1 expression was not significantly related to PD-L1 and FOXP3 expression (Figure 3I,J), whereas low Fh1 expression was related to decreased CD8 expression (Figure 3K). These findings suggest that knockdown of FH inhibits CD8+ T cell clonal expansion in CRC and reduces the therapeutic effect of PD-1 antibodies. The development of new molecular entities is helpful in improving the effect of immunotherapy [12,13]. In the future, drugs related to FH might be beneficial to improve the prognosis of patients with CRC, which is worthy of further exploration.

### 3.4. FH Negatively Regulates PCSK9 Expression in CRC

In order to explore how FH regulates CD8+ T cell clonal expansion and thus affects the therapeutic effect of PD-1 antibodies, we investigated the downstream effectors of FH and compared the differentially expressed genes between FH knockdown and control cells using RNA-seq (GSE245475). Our analysis showed that PRSS1, PRSS2, and PCSK9 were significantly upregulated in CRC cells with FH knockdown (Figure 4A). Because the previous literature has proven the influence of PCSK9 on CD8+ T cell clonal expansion and PD-1 antibody treatment [10,17] well, but there is no such report on PRSS1 and PRSS2, we chose PCSK9 for the next research. To confirm the relation between FH and PCSK9, we generated FH stable knockdown/overexpression cell lines. We observed that FH knockdown promoted PCSK9 mRNA and protein expression (Figure 4B,C). Furthermore, overexpression of FH reduced PCSK9 mRNA and protein expression (Figure 4D,E). Previous studies have shown that the epigenetic regulation of FH on downstream genes is achieved through fumaric acid, which is the methyl donor of epigenetic modification of DNA [18]. As low expression of FH can cause an increase in fumarate (FA), we performed fumarate gradient experiments in SW620 cells without/with knockdown of FH. The results showed a slight but not statistically significant change in PCSK9 mRNA levels with increasing fumarate concentration (Appendix A). Therefore, FH does not regulate the expression of PCSK9 through fumaric acid. We hypothesize that FH may directly affect the nuclear localization of the PCSK9 transcription factor and then regulate the expression of PCSK9. Therefore, we further verified the relation between FH and PCSK9 expression by immunohistochemistry in CRC tissue microarrays, which showed that FH was inversely related to PCSK9 expression (Figure 4F,G). Therefore, we constructed a cell line with stable knockdown of FH and PCSK9 using HCT116 cells and established subcutaneous xenograft nude mice to observe their effects on tumor growth. The in vivo experiments showed that the knockdown of PCSK9 can inhibit the growth of tumors, while the knockdown of FH seems to weaken this effect (Figure 4H). These findings suggest that PCSK9 is a downstream component of FH and that FH can negatively regulate PCSK9 expression. Based on the currently reported effects of PCSK9 on immunotherapy [10], we hypothesize that FH may modulate CD8+ T cell clonal expansion through PCSK9.

### 3.5. FH Inhibits the Nuclear Translocation of the PCSK9 Transcription Factor SREBF1/2

Considering that FH affects the mRNA and protein levels of PCSK9, we hypothesize that FH regulates the transcription process of PCSK9. To elucidate the mechanisms underlying the transcriptional regulation of the PCSK9 gene, we identified four major transcription factors related to PCSK9 by enrichment analysis in the ChIP-Atlas (http://chip-atlas.org/, accessed on 12 June 2023) database (Figure 5A), which were consistent with previous reports [19,20,21]. We obtained the subcellular localization of these four transcription factors using Uniport (http://www.uniprot.org/, accessed on 12 June 2023) and found that both SREBF1 and SREBF2 were located in the nucleus and cytoplasm, whereas HNF1A and HNF1B were mainly located in the nucleus (Figure 5B). We next quantified the PCSK9 promoter DNA fragments in FH-knockdown cells using ChIP-qPCR. Our results showed that only SREBF1 and SREBF2 showed increased binding to the PCSK9 promoter after knockdown of FH (Figure 5C). This suggests that SREBF1 and SREBF2 may play a key role in the regulation of PCSK9 by FH. We hypothesize that the regulatory effects of SREBF1 and SREBF2 on PCSK9 may be related to nuclear translocation. Therefore, we used immunofluorescence to analyze the expression and localization of SREBF1 and SREBF2 in cells with FH knockdown and FH overexpression further. We established stable cell lines using SW620 and HCT116 with FH overexpression or knockdown and found that the localization of SREBF1 and SREBF2 in the nucleus and cytoplasm was altered (Figure 5D). The changes in SREBF1 and SREBF2 expression in the nucleus and cytoplasm were examined by Western blot. Our results showed that the expression of SREBF1 and SREBF2 in the nucleus was significantly increased when FH was knocked down and decreased when FH was overexpressed (Figure 5E). Thus, the function of SREBF1 and SREBF2 may be dependent on subcellular localization. These results suggest that FH may regulate the expression of PCSK9 by inhibiting the nuclear translocation of SREBF1/2.

We further investigated the potential effect of FH expression on the nuclear translocation and expression of SREBF1 and SREBF2 is not by fumaric acid (FA). The results showed that the presence of FA did not affect the expression of SREBF1 and SREBF2 in the nucleus or cytoplasm when FH was knocked down. Therefore, we used immunofluorescence to analyze the expression and localization of SREBF1 and SREBF2 in FA-knockdown cells further, which suggested that FA has no effect on the nuclear translocation of SREBF1 and SREBF2. Thus, FH may not inhibit the nuclear translocation of SREBF1/2 through FA (Appendix A).

### 3.6. FH Regulates the Immune Response in CRC Cells through the RAN-SREBF1/2-PCSK9 Signal Axis

Nuclear translocation of SREBF1/2 requires passage through Ran-GTP [22,23]. The STRING database was used to predict interacting proteins with FH, and label-free quantitative proteomic analysis of FH interacting proteins was performed using protein mass spectrometry. We found that Ran protein may be a regulator between FH and transcription factors (Figure 6A,B). Therefore, we used Co-IP to analyze the interaction between FH, Ran, SREBF1, SREBF2, HNF1A, and HNF1B in HCT116 cells. However, our results indicate that FH does not directly bind to SREBF1, SREBF2, HNF1A, and HNF1B (Figure 6C). We found that Ran binds directly to the SREBF1 and SREBF2 proteins (Figure 6D). Further experiments demonstrated that FH directly binds to the Ran protein, which only binds to SREBF1 and SREBF2 proteins but not to HNF1A and HNF1B. In addition, FH does not directly bind to PCSK9 (Figure 6E). When FH was knocked down, the binding of Ran protein to SREBF1 and SREBF2 was significantly increased (Figure 6F). These results suggest that the binding of FH to Ran inhibits the nuclear translocation of SREBF1 and SREBF2. PCSK9 can regulate the clonal expansion of cytotoxic T cells by affecting major histocompatibility protein class I (MHC I) proteins [10]. Our results showed that when FH and Ran were knocked down, the expression of PCSK9 decreased, and the expression of HLA-A increased. When FH was overexpressed and Ran was knocked down, the expression of PCSK9 decreased, and the expression of HLA-A increased (Figure 6G). Based on these findings, we hypothesized that PCSK9 may attenuate the immunotherapeutic effect when FH is lowly expressed. Consequently, the use of PCSK9 inhibitors may be able to enhance the immunotherapeutic effect. Next, we selected the MC38 cell line to construct an Fh1 stable knockdown expression cell line and established subcutaneous CRC tumor-bearing C57BL/6 mice to observe the effect of shFh1, αPD-1 and PCSK9i on tumor growth. Our in vivo experiments showed that the knockdown of FH affected the volume and magnitude of tumor regression after PD-1 immunotherapy and PCSK9 inhibition reversed the effect of immunotherapy (Figure 6H). Taken together, these findings suggest that low expression of FH may weaken the immunotherapeutic effect, which may be enhanced by PD-1 antibodies and PCSK9 inhibitors combination therapy.

## 4. Discussion

Immunosuppressive regulation of the tumor microenvironment is a promising therapeutic approach for the treatment of various malignant tumors [24]. However, due to the poor infiltration of cytotoxic T lymphocytes (CTL), the effect of immunotherapy in CRC tumors with pMMR and, to a lesser extent, dMMR is limited [5,6]. Our findings provide new insights for improving the effect of CRC immunotherapy. We observed that low FH expression was related to the insensitivity of immunotherapy in patients with CRC. Mechanistically, decreased FH expression promotes Ran-mediated SREBF1/2 nuclear translocation, leading to increased PCSK9 expression, which in turn leads to decreased CD8+ T cell clonal expansion and ultimately weakens the effect of PD-1 antibodies monotherapy. Therefore, FH may be a promising biomarker for PD-1 antibody treatment in CRC.

Metabolic alterations involved in tumorigenesis have multifaceted effects, not only relating to the nutritional status of cells but also influencing cell behavior, including the regulation of signaling and mechanics [25]. As a key metabolic enzyme in the TCA cycle, FH loss caused by mutation or transcriptional inhibition is related to the occurrence of a variety of cancers [26]. Dysregulation of energy homeostasis resulting from FH loss leads to the activation of key oncogenic pathways and transcriptional programs, such as those regulated by HIF, mTOR, and PI3K [16]. Furthermore, depletion of FH and subsequent accumulation of FA can induce EMT, thereby promoting aggressive tumor characteristics [18]. Recent studies have demonstrated that dysfunction of the FH gene may adversely impact relapse-free survival and overall survival rates in patients with CRC [15]. Our findings reveal a significant downregulation of FH expression in CRC tissues compared with normal tissues. Moreover, low expression of FH is positively related to poor prognosis in patients with CRC. Additionally, FH plays a pivotal role in CRC cell proliferation and invasion. Low expression of FH impedes CD8+ T-cell clonal expansion and diminishes the therapeutic efficacy of PD-1 antibodies. These results underscore the crucial involvement of FH in CRC and its potential as a promising biomarker for therapeutic intervention.

Our findings indicate that PCSK9 is a downstream target of FH, and FH can negatively regulate the expression of PCSK9 in CRC. Previous studies have demonstrated that the deletion or pharmacological inhibition of PCSK9 in tumor cells can enhance the anti-tumor activity of CD8+ T cells, subsequently impeding tumor progression [11]. Simultaneously, PCSK9 plays a crucial role in regulating MHC I levels on cell surfaces, influencing immune infiltration within tumors, and responding to immune checkpoint therapy [10]. Our findings corroborate previous research and provide additional evidence for the involvement of PCSK9 in immune microenvironment modulation and immunotherapy. These results reveal novel regulatory mechanisms of PCSK9 from the perspective of metabolic reprogramming and lay a solid foundation for future interventional applications of PCSK9 as an immunotherapeutic target.

The nuclear translocation of SREBF1/2 mediated by Ran protein is one of the transcription regulation mechanisms that affect the expression of PCSK9 [21]. Both in vitro and in vivo transcriptional levels are regulated by SREBPs, with SREBP-2 being the primary regulator responsible for sterol-dependent PCSK9 expression in vivo [19,20]. Furthermore, ran proteins facilitate the nuclear translocation of PCSK9. Ran GTPases belong to the Ras superfamily and govern nucleoplasmic transport through nuclear pore complexes while also regulating microtubule polymerization and mitotic spindle formation to control cell cycle progression [23,27]. Disruption of Ran expression is implicated across various stages of cancer development, from carcinogenesis to metastasis [28,29]. Our findings suggest that SREBF1/2 acts as a transcription factor for PCSK9. Additionally, FH binds to Ran and influences SREBF1/2 nuclear import, resulting in decreased PCSK9 expression. Conversely, knockdown of FH enhances SREBF1/2 nuclear import and upregulates PCSK9 expression. In addition, there is no doubt that the process of CRC progression and treatment resistance is complicated. More detailed relevant mechanisms of FH are indeed worthy of further exploration in the future.

Despite remarkable advancements in immunotherapy, a subset of patients with CRC fail to respond due to limited infiltration of T lymphocytes [30]. Consequently, there is growing interest in developing rational combinations that can attract T lymphocytes into these tumors. Preclinical and clinical studies have demonstrated efficacy combined with radiation, high-dose IL-2, and IL-10 through the expansion of CD8+ T cells against melanoma and renal cell carcinoma [31,32]. However, this therapeutic effect relies on continuous administration of high doses of drugs, which lead to toxicity concerns and hamper the development of tumor immunotherapy combinations. Our findings reveal that combining a PCSK9 inhibitor with PD-1 antibodies promotes CD8+ T cell clonal expansion and significantly enhances tumor immunotherapy in an FH-low expression mouse model of CRC. Considering these results, our findings hold potential for future clinical applications. It is undeniable that the evidence for clinical validation of our findings is limited. Therefore, these findings need to be verified in a larger cohort in the future. Further prospective recruitment research samples were evaluated.

## 5. Conclusions

In conclusion, our findings suggest a positive relation between low FH expression and poor prognosis, as well as reduced CD8+ T cell clonal expansion in patients with CRC. The downregulation of FH expression promotes the nuclear translocation of Ran-mediated SREBF1/2, resulting in increased PCSK9 expression. This leads to decreased CD8+ T cell clonal expansion and ultimately weakens the efficacy of PD-1 antibody monotherapy (Figure 7). Therefore, combined therapy targeting PCSK9 and PD-1 may be beneficial for CRC patients with low FH expression.

## Figures and Tables

**Figure 1 cancers-16-00713-f001:**
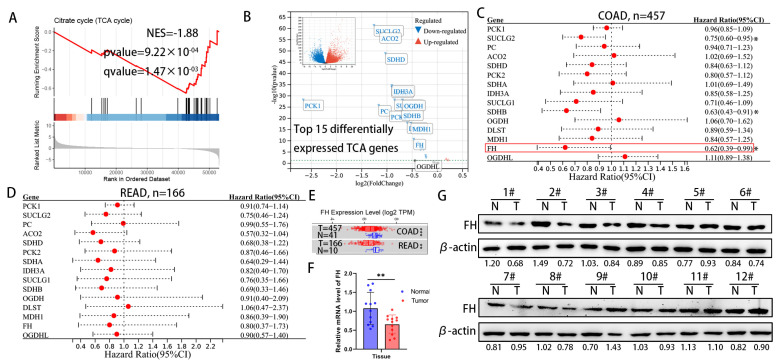
Downregulation of FH expression in CRC predicts poor prognosis. (**A**) GSEA results showed significant enrichment of the citric acid cycle (TCA cycle) pathway in normal tissues. (**B**) Differentially expressed genes were displayed using volcano plots, showing the top 15 significantly differentially expressed TCA genes. (**C**) Hazard ratios and 95% confidence intervals for each gene of colon adenocarcinoma (COAD) are shown in forest plots. (**D**) Hazard ratios and 95% confidence intervals for each gene rectal adenocarcinoma (READ) are shown in forest plots. (**E**) FH expression levels in normal and tumor tissues of COAD and READ. (**F**) The mRNA levels of FH in 12 pairs of CRC tissues were detected by RT-qPCR. (**G**) The protein levels of FH in 12 pairs of CRC tissues were detected by Western blotting. The uncropped bolts are shown in Appendix A. T, CRC tissue, N, adjacent nontumor tissue. Relative * *p* < 0.05; ** *p* < 0.01; *** *p* < 0.001.

**Figure 2 cancers-16-00713-f002:**
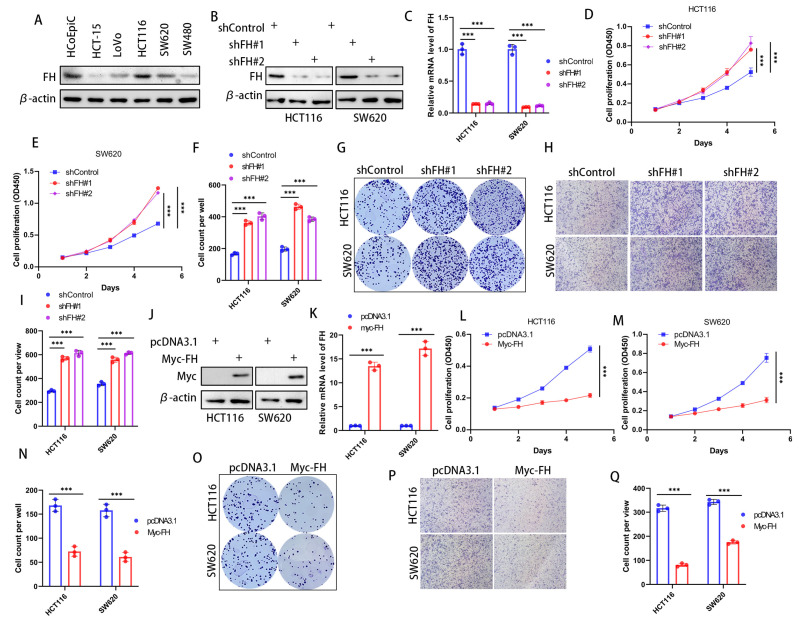
FH inhibits the proliferation and invasion of CRC cells. (**A**) FH expression in a normal colonic epithelial cell line HCoEpiC and five different CRC cell lines. (**B**) Transfection and knockdown efficiencies were evaluated by Western blotting. (**C**) RT-qPCR was used to determine the transfection efficiency of gene knockdown. (**D**,**E**) Growth curves of HCT116 and SW620 cells after FH knockdown were plotted using CCK-8 analysis (*n* = 3). (**F**,**G**) Colony formation assay was used to detect the colony-forming ability of HCT116 and SW620 cells after overexpression of FH. (**H**,**I**) The invasive ability of HCT116 and SW620 cells after FH knockdown was determined by Transwell assay (*n* = 3). (**J**) Transfection and overexpression efficiencies were assessed by Western blotting. (**K**) The transfection efficiency of gene overexpression was determined using RT-qPCR. (**L**) Growth curves of HCT116 cells after FH overexpression were plotted using CCK-8 assay (*n* = 3). (**M**) Growth curves of SW620 cells after FH overexpression were plotted using CCK-8 assay (*n* = 3). (**N**,**O**) Colony formation assay was used to analyze colony-forming ability. (**P**,**Q**) The invasive ability of HCT116 and SW620 cells after overexpression of FH was determined by transwell assay (*n* = 3). Relative *** *p* < 0.001. The uncropped bolts are shown in Appendix A.

**Figure 3 cancers-16-00713-f003:**
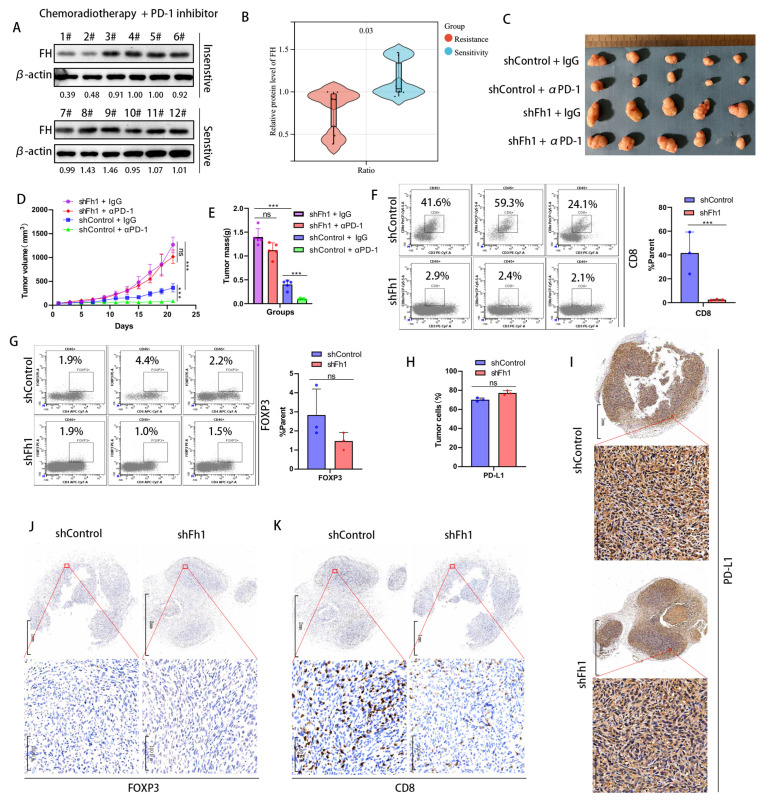
Low expression of FH inhibits the clonal expansion of CD8+ T cells in CRC and reduces the therapeutic effect of PD-1 antibodies. (**A**,**B**) Western blot analysis of FH expression in patients sensitive (*n* = 6) or insensitive (*n* = 6) to chemotherapy plus PD-1 antibodies. (**C**) Subcutaneous implantation of tumors in mice (*n* = 5) after infection of MC38 cells with the corresponding lentiviral particles. (**D**,**E**) Mean growth curves and tumor weights (*n* = 5) of subcutaneously transplanted tumors from mice infected with MC38 cells by the corresponding lentiviral particles. (**F**) Proportion of CD8+ T cells in subcutaneously transplanted tumors of mice infected with MC38 cells by corresponding lentivirus particles. (**G**) The proportion of Treg cells in the subcutaneous xenograft tumor of the corresponding lentiviral particles infected with MC38 cells (**H**) The proportion of PD-L1-positive tumor cells in the tumor tissue of the shControl group and the shFh1 group. (**I**) Representative IHC staining for PD-L1 in tumor tissue sections of shControl and shFh1 mice. (**J**) Representative IHC staining of FOXP3 in tumor tissue sections from shControl and shFh1 mice. (**K**) Representative IHC staining for CD8 in tumor tissue sections from shControl and shFh1 mice. Relative ns, no significance; *** *p* < 0.001. The uncropped bolts are shown in Appendix A.

**Figure 4 cancers-16-00713-f004:**
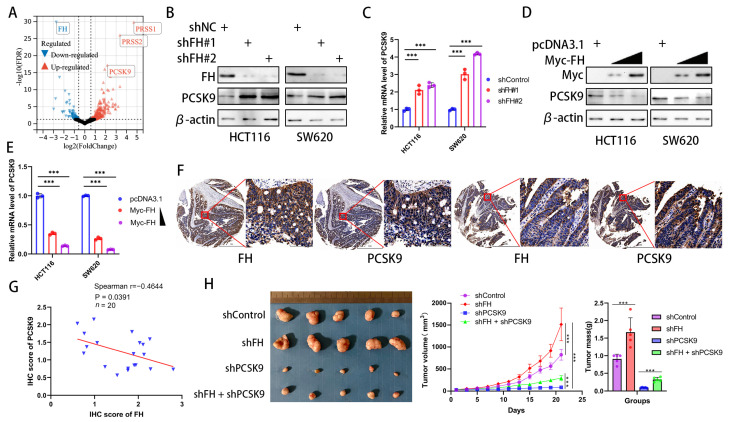
FH negatively regulates PCSK9 expression in CRC. (**A**) Comparison of the RNA-seq volcano plot results of CRC cells from shControl and shFH. (**B**) The protein levels of FH and PCSK9 in HCT116 and SW620 cells with FH knockdown were detected by Western blotting. (**C**) The mRNA levels of FH and PCSK9 in HCT116 and SW620 cells with FH knockdown were detected by RT-qPCR. (**D**) The protein levels of FH and PCSK9 in HCT116 and SW620 cells overexpressing FH were detected by Western blotting. (**E**) The mRNA levels of PCSK9 in HCT116 and SW620 cells with FH overexpression were detected by RT-qPCR. (**F**) Immunostaining of FH and PCSK9 in tumors from representative human CRC cases with different FH expression. Scale bars: 1 mm (left), 100 μm (right). (**G**) Linear regression relation between IHC score for PCSK9 in CRC and IHC score for FH. The blue triangle represents the IHC score of FH and PCSK9. (**H**) Images of transplanted tumors in nude mice infected with the corresponding lentivirus particles, as well as the average growth curve and tumor weight (*n* = 5). Relative *** *p* < 0.001. The uncropped bolts are shown in Appendix A.

**Figure 5 cancers-16-00713-f005:**
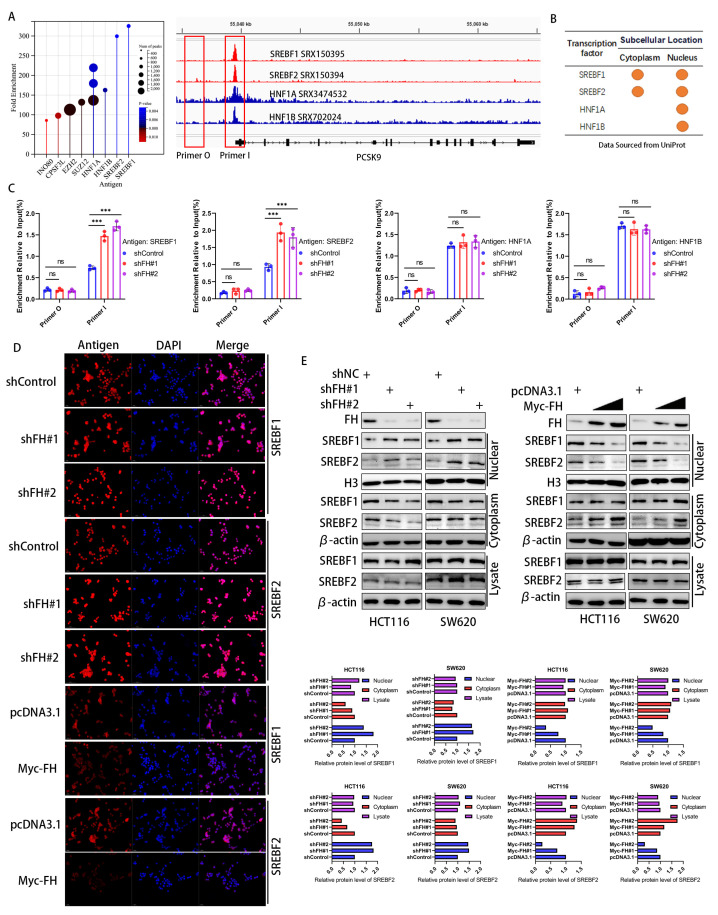
FH inhibits the nuclear translocation of the PCSK9 transcription factor SREBF1/2. (**A**) Screening and identification of potential PCSK9 transcription factors based on ChIP-seq peaks in gene promoters. (**B**) Subcellular location of proteins by Uniprot. (**C**) ChIP-qPCR assay was used to evaluate the DNA fragmentation quantification of the transcription factors SREBF1, SREBF2, HNF1A, and HNF1B in the promoter region after FH knockdown. (**D**) After FH knockdown or overexpression in SW620 cells, immunofluorescence was used to locate SREBF1, and SREBF2 proteins. (**E**) Western blot was used to analyze the expression levels of SREBF1 and SREBF2 in the nucleus and cytoplasm after FH knockdown or overexpression. Relative ns, no significance; *** *p* < 0.001. The uncropped bolts are shown in Appendix A.

**Figure 6 cancers-16-00713-f006:**
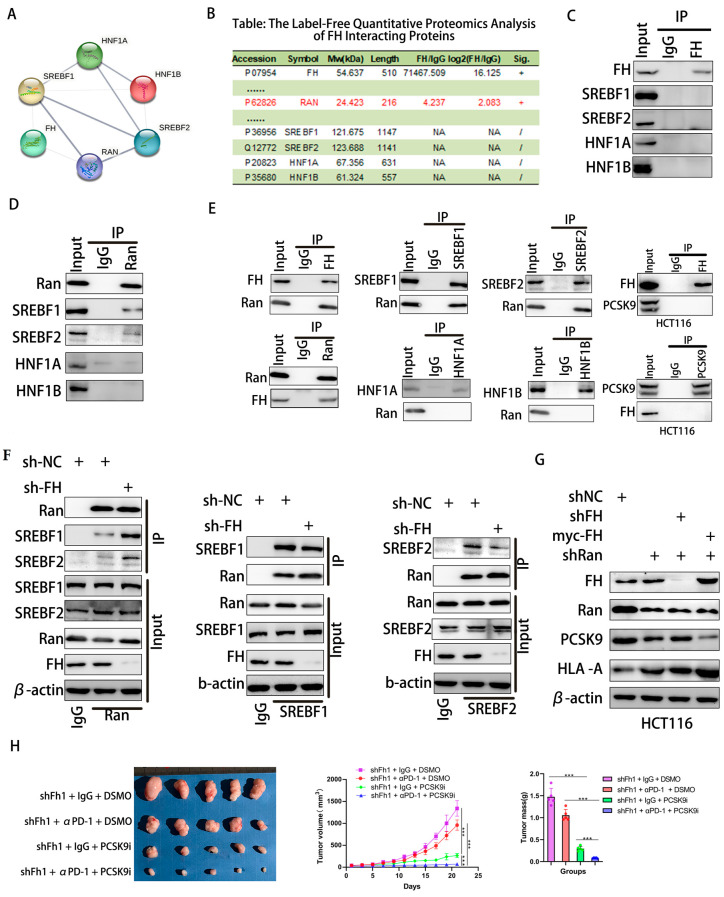
FH regulates the immunotherapy response in CRC cells through the RAN-SREBF1/2-PCSK9 signaling axis. (**A**,**B**) PPI networks of FH and label-free quantitative proteomic analysis of FH-interacting proteins. (**C**) Analysis of the interaction between FH and SREBF1, SREBF2, HNF1A, and HNF1B in HCT116 cells using Co-IP. (**D**) Co-IP analysis of the interaction between Ran and SREBF1, SREBF2, HNF1A, and HNF1B in HCT116 cells. (**E**) Co-IP was performed in HCT116 cells using FH, SREBF1, SREBF2, Ran, PCSK9, HNF1A and HNF1B antibodies. (**F**) Co-IP analysis of Ran, SREBF1, and SREBF2 binding in FH knockdown HCT116 cells. (**G**) Analysis of FH, Ran, PCSK9, and HLA-A using Western blot after knockdown of FH, Ran, and/or overexpression of FH. (**H**) Images of subcutaneous transplanted tumors in mice infected with Fh1 lentivirus particles and the average growth curve and tumor weight (*n* = 5). Relative ns, no significance; *** *p* < 0.001. The uncropped bolts are shown in Appendix A.

**Figure 7 cancers-16-00713-f007:**
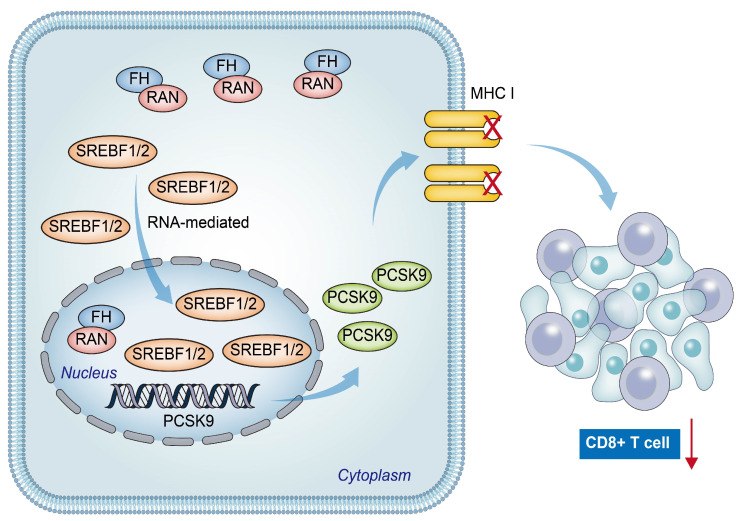
The hypothetical model depicts that reduced FH expression promotes Ran-mediated nuclear translocation of SREBF1/2, leading to increased PCSK9 expression and eventual suppression of the immunotherapy response in colorectal cancer. The red arrow represents the reduction of CD8+ T cell clonal expansion.

## Data Availability

Data are contained within the article and Appendix A.

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
