# Peer review of "Fumarate Hydratase Enhances the Therapeutic Effect of PD-1 Antibody in Colorectal Cancer by Regulating PCSK9"

_cancers, 2024, doi:10.3390/cancers16040713_

Round 1
Reviewer 1 Report
Comments and Suggestions for Authors
This study by Qin et al addresses the limited efficacy of PD-1 antibodies in treating CRC due to metabolic reprogramming that affects the tricarboxylic acid cycle. Specifically, it identifies the down-regulation of FH as a factor in poor CRC prognosis. Using a CRC mouse model with deficient Fh1 expression demonstrates that PD-1 antibody treatment is less effective in such cases, but combining it with PCSK9 inhibitors improves outcomes. The mechanism involves FH's interaction with RAN, influencing PCSK9 expression and thus affecting CD8+T cell infiltration and the immunotherapy response. This suggests that CRC patients with low FH expression might benefit more from a combination of PD-1 antibodies and PCSK9 inhibitors.
- The study highlights a potential therapeutic strategy for CRC patients unresponsive to PD-1 antibodies. However, its scope is limited to preclinical models, and translating these findings to clinical practice requires further validation.
- The study focuses narrowly on one metabolic pathway and its impact on immunotherapy, possibly overlooking other relevant mechanisms in CRC progression and treatment resistance.
- Before concluding, it is crucial to acknowledge the limitations of this study.
- Please add line numbering to facilitate the peer review process.
- the figures are well presented
Well done
Author Response
1.The study highlights a potential therapeutic strategy for CRC patients unresponsive to PD-1 antibodies. However, its scope is limited to preclinical models, and translating these findings to clinical practice requires further validation.
Response: Thanks for your helpful advice. The aim of our study was mainly to provide evidence support for the protective role of FH in CRC from a basic research perspective. It is undeniable that the evidence for clinical validation of our findings is limited. Therefore, these findings need to be verified in a larger cohort in the future. And further prospective recruitment research samples were evaluated. We have supplemented the limitations of this study in manuscript (page 20, line 31). Changes to the manuscript are shown in the text in red.
2.The study focuses narrowly on one metabolic pathway and its impact on immunotherapy, possibly overlooking other relevant mechanisms in CRC progression and treatment resistance.
Response: Thanks for your helpful advice. There is no doubt that the process of CRC progression and treatment resistance is complicated. Our study focused on genes affecting the TCA cycle and identified the potential clinical value of FH. We verified the possible mechanism of FH. Mechanistically, FH binds to RAN, which inhibits the nuclear import of the PCSK9 transcription factor, SREBF1/2, thus reducing the expression of PCSK9. More detailed relevant mechanisms are indeed worthy of our further exploration in the future. This is a limitation of our study, and we have already addressed this limitation in the manuscript (page 20, line 17). Changes to the manuscript are shown in the text in red.
3.Before concluding, it is crucial to acknowledge the limitations of this study.
Response: Thanks for your helpful advice. We have described the limitations of this study (page 20). Changes to the manuscript are shown in the text in red.
4.Please add line numbering to facilitate the peer review process.
Response: Thanks for your helpful advice. This change is important for a more facilitated peer review process. We have added the line numbers for each page.
5.the figures are well presented
Response: Thank you for your endorsement. There is no doubt that these suggestions have made a substantial improvement in our article. If there are any questions you have about our submission, please feel free to contact me.

Reviewer 2 Report
Comments and Suggestions for Authors The manuscript cancers-2799512 entitled "Fumarate hydratase enhances the therapeutic effect of PD-1 antibody in colorectal cancer by regulating PCSK9" is a well performed study highlighting a new scenario for CRC treatment. the work is well conducted and experiments confirm the hypotheses. I think that the manuscript merits publication but some points must be revised, in particular I have a question:ù is this approach (drug targeting) useful for the development of new molecular entities able to interfere with this pathway and revert colon cancer aggressiveness? Please discuss this hypothesis in the introduction and results section.Author Response
The manuscript cancers-2799512 entitled "Fumarate hydratase enhances the therapeutic effect of PD-1 antibody in colorectal cancer by regulating PCSK9" is a well performed study highlighting a new scenario for CRC treatment. the work is well conducted and experiments confirm the hypotheses. I think that the manuscript merits publication but some points must be revised, in particular I have a question:ù is this approach (drug targeting) useful for the development of new molecular entities able to interfere with this pathway and revert colon cancer aggressiveness? Please discuss this hypothesis in the introduction and results section.
Response: Thanks for your helpful advice. Our main research result is that FH has a protective effect on the prognosis of CRC. In addition, CRC patients with low FH expression may benefit from combinatorial therapy with PD-1 antibodies and PCSK9 inhibitors to enhance curative effect. Of course, if we can design a new molecular entities to revert the invasion of colorectal cancer by regulating FH, it is also an interesting study. The development of new molecular entities is helpful to improve the effect of immunotherapy [1, 2]. In the future, drugs related to FH might be beneficial to improve the prognosis of patients with CRC, which is worthy of further exploration. Thank you for your exploratory suggestion. We have discussed this hypothesis in the introduction (page 3, line 11) and discussion section (page 11, line 28). There is no doubt that these suggestions have made a substantial improvement in our article. If there are any questions you have about our submission, please feel free to contact me.
[1].Tapia-Galisteo A, Compte M, Álvarez-Vallina L, Sanz L. When three is not a crowd: trispecific antibodies for enhanced cancer immunotherapy. Theranostics. 2023;13(3):1028-1041. Published 2023 Jan 22.
[2].Masson C, Thouvenin J, Boudier P, et al. Biological Biomarkers of Response and Resistance to Immune Checkpoint Inhibitors in Renal Cell Carcinoma. Cancers (Basel). 2023;15(12):3159. Published 2023 Jun 12.
Reviewer 3 Report
Comments and Suggestions for Authors
This work elegantly demonstrates a mechanistic role of FH in tumor immune evasion in CRC patients. While the experimental work is very impressive and I commend the authors for their detailed approach, however authors should further address the following comments.
1. Figure 1F: authors should mention which patient samples were insensitive to therapy. Also, an mRNA expression data would be useful.
2. Figure 2: authors should also perform some recue experiments in knockdown cells, to show the dominant effect of FH silencing on cell proliferation/viability? siRNA effects are transient and may not represent the true directionality.
3. Figure 3F, shows knockdown of FH inhibits CD8+T cell infiltration to tumors. Is this also related to low migration of CD8 T cells to tumor site/ TME? Still the question remains, how loss of FH expression will alter priming of CD8 T cells to TME?
4. Figure 4 & 5: authors speculate that FH regulates the transcription process of PCSK9. Does this also related to the possibility of epigenetic regulation by FH or PCSK9 to each other.
5. Does FH and PCSK9 interacts at protein level? Any co-immunoprecipitation data?
6. I think it would be beneficial to provide graphical abstract or summary model to explain the study.
Minor comments:
1. Rephrase this sentence, it’s not clear, page 2: “and low expression of FH caused by deletion or transcription inhibition is relatived with cancer”.
2. Page 2: need better integration between FH and PCSK9.
3. I would suggest authors to change word speculate to hypothesize.
Comments on the Quality of English LanguageNeed improvement.
Author Response
This work elegantly demonstrates a mechanistic role of FH in tumor immune evasion in CRC patients. While the experimental work is very impressive and I commend the authors for their detailed approach, however authors should further address the following comments.
1.Figure 1F: authors should mention which patient samples were insensitive to therapy. Also, an mRNA expression data would be useful.
Response: Thanks for your helpful advice. In fact, the main purpose of this batch of patient specimens is to verify the expression of FH in cancer and normal tissues. Therefore, not every patient has received treatment, so there is a lack of complete data on whether treatment is sensitive. For the evidence on whether FH affects the response to immunotherapy, we have shown it in the Figure 3A. As to the mRNA expression of FH in cancer and normal tissues, your suggestion was very rigorous, so we have supplemented the data in this section (Figure 1G). Changes to the manuscript are shown in the text in red.
2.Figure 2: authors should also perform some recue experiments in knockdown cells, to show the dominant effect of FH silencing on cell proliferation/viability? siRNA effects are transient and may not represent the true directionality.
Response: Thanks for your helpful advice. We supplemented this section with rescue experiments, and the results still support our original conclusion. The proliferation/viability ability of cancer cells was significantly elevated in the presence of low FH expression. After overexpressing FH in cancer cells, the proliferation/viability ability of cancer cells caused by low FH expression were inhibited (Figure S3C). Changes to the manuscript are shown in the text in red.
3.Figure 3F, shows knockdown of FH inhibits CD8+T cell infiltration to tumors. Is this also related to low migration of CD8 T cells to tumor site/ TME? Still the question remains, how loss of FH expression will alter priming of CD8 T cells to TME?
Response: Thanks for your helpful advice. For this part, our expression may not be clear. Low-density lipoprotein receptor (LDLR) is essential for the CD8+ T cell priming, clonal expansion, and effector function. Moreover, PCSK9 can bind to LDLR and prevent the recycling of LDLR and TCR to the plasma membrane [1]. Therefore, considering that our results show the regulatory effect of FH on PCSK9, FH can enhance the anti-tumor activity of CD8+T cells by inhibiting PCSK9.
[1]. Yuan J, Cai T, Zheng X, et al. Potentiating CD8+ T cell antitumor activity by inhibiting PCSK9 to promote LDLR-mediated TCR recycling and signaling [published correction appears in Protein Cell. 2022 Sep;13(9):694-700]. Protein Cell. 2021;12(4):240-260.
4.Figure 4 & 5: authors speculate that FH regulates the transcription process of PCSK9. Does this also related to the possibility of epigenetic regulation by FH or PCSK9 to each other.
Response: Thanks for your helpful advice. Previous studies have shown that the epigenetic regulation of FH on downstream genes is achieved through fumaric acid, which is the methyl donor of epigenetic modification of DNA [1]. As low expression of FH can cause an increase in fumarate, we performed fumarate gradient experiments in SW620 cells without/with knockdown of FH. The results showed a slight but not statistically significant change in PCSK9 mRNA levels with increasing fumarate concentration. Therefore, FH does not regulate the expression of PCSK9 through fumaric acid. We hypothesize that FH may directly affect the nuclear localization of PCSK9 transcription factor, and then regulate the expression of PCSK9. This question have been explained in the manuscript (page 13, line 14). Changes to the manuscript are shown in the text in red.
[1]. Sciacovelli M, Gonçalves E, Johnson TI, et al. Fumarate is an epigenetic modifier that elicits epithelial-to-mesenchymal transition [published correction appears in Nature. 2016 Dec 1;540(7631):150]. Nature. 2016;537(7621):544-547.
5.Does FH and PCSK9 interacts at protein level? Any co-immunoprecipitation data?
Response: Thanks for your helpful advice. Your considerations are very rigorous and we have supplemented this Co-IP experiment. And the results show that FH does not interact directly with PCSK9 (Figure 6E). Therefore, FH affects PCSK through a series of mechanisms, that is, FH binds to RAN, which inhibits the nuclear import of the PCSK9 transcription factor, SREBF1/2, thus reducing the expression of PCSK9.
6.I think it would be beneficial to provide graphical abstract or summary model to explain the study.
Response: Thanks for your helpful advice. In Figure 7 (page 21), we have showed the model diagram of this study.
Minor comments:
7.Rephrase this sentence, it’s not clear, page 2: “and low expression of FH caused by deletion or transcription inhibition is relatived with cancer”.
Response: Thanks for your helpful advice. FH is an important metabolic enzyme in the TCA cycle, and the loss of FH can act as cancer driver. We have perfected the content of this part in manuscript (page 2, line 28). Changes to the manuscript are shown in the text in red.
8.Page 2: need better integration between FH and PCSK9.
Response: Thanks for your helpful advice. Through RNA-seq, we found that PCSK9 may be a downstream effector of FH. This section has been modified in manuscript (page 2, line 32). Changes to the manuscript are shown in the text in red.
9.I would suggest authors to change word speculate to hypothesize.
Response: Thanks for your helpful advice. We have modified this word. Changes to the manuscript are shown in the text in red. There is no doubt that these suggestions have made a substantial improvement in our article. If there are any questions you have about our submission, please feel free to contact me.
Round 2
Reviewer 3 Report
Comments and Suggestions for Authors
Authors have addressed all the comments, I have no further comments.